# Is a Miracle-less WIMP Ruled out?

Jason Arakawa , Tim M.P. Tait

Department of Physics and Astronomy, University of California, Irvine, CA 92697-4575 USA

## Abstract

We examine a real electroweak triplet scalar field as dark matter, abandoning the requirement that its relic abundance is determined through freeze out in a standard cosmological history (a situation which we refer to as 'miracle-less WIMP'). We extract the bounds on such a particle from collider searches, searches for direct scattering with terrestrial targets, and searches for the indirect products of annihilation. Each type of search provides complementary information, and each is most effective in a different region of parameter space. LHC searches tend to be highly dependent on the mass of the SU(2) charged partner state, and are effective for very large or very tiny mass splitting between it and the neutral dark matter component. Direct searches are very effective at bounding the Higgs portal coupling, but ineffective once it falls below $\lambda_{\text{eff}} \lesssim 10^{-3}$. Indirect searches suffer from large astrophysical uncertainties due to the backgrounds and $J$-factors, but do provide key information for $\sim 100$ GeV to TeV masses. Synthesizing the allowed parameter space, this example of WIMP dark matter remains viable, but only in miracle-less regimes.

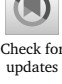

# 1 Introduction

The nature of dark matter has persisted as one of the most vital open questions necessary for our understanding the universe's fundamental building blocks. The Standard Model (SM) does not incorporate dark matter, and other unsolved problems within the SM could point towards clues to its nature. Many theories of physics beyond the standard model (BSM) predict new fields with roughly electroweak-sized interactions and masses, some of which also have the correct properties to be the dark matter. These candidates, called weakly interacting massive particles (WIMPs), are among the most compelling and well-studied, largely because the freeze out mechanism naturally suggests a relic abundance similar to the one inferred from cosmological measurements [1,2]. The typical story assumes a standard cosmological history which extrapolates the SM back into the early universe, leading to the dark matter being in thermal equilibrium with the SM particles at early times. As the temperature of the Universe falls, the dark matter's interactions eventually freeze out, resulting in a fixed comoving density. For particles such as WIMPs, the abundance of dark matter derived from freeze out roughly corresponds to the observed abundance, a coincidence that is often referred to as the 'WIMP Miracle'.

Despite this attractive picture, there is a growing sense that WIMPs are no longer favored as a candidate to play the role of dark matter. The null results from direct and indirect searches for dark matter in the Galaxy and for its production at colliders have ruled out portions of the parameter space living at the heart of the WIMP miracle. While a large part of this shift in focus is simply driven by the healthy urge to explore a wider parameter space [3] particularly since no concrete observation suggests that the WIMP miracle is realized in nature, it remains important to map out the boundary between what types of WIMP dark matter are allowed, and which are concretely ruled out by null searches. Even in the context of a standard cosmology, WIMP-like dark matter particles whose relic abundance is determined by freeze out remain viable for a range of parameter space (see e.g. [4,5]).

In this work, we explore a related but distinct question, regarding the viability of a dark matter particle with standard electroweak interactions, but whose relic abundance is not set by freeze out during a standard cosmology. Given the lack of solid observational probes at times before Big Bang Nucleosynthesis (BBN), which itself occurs long after a typical WIMP would have frozen out, it is not difficult to imagine modifications to the standard cosmology which are consistent with observations, but yield a radically different picture of the parameter space favored by its abundance [6–9]. We focus on the simple representative case of dark matter described by a real scalar field transforming as a triplet under the $SU(2)_{EW}$ interaction of the Standard Model. This construction was previously considered as a specific case of "Minimal Dark Matter" in Ref. [10] and (aside from spin) is similar to the limit of the Minimal Supersymmetric Standard Model in which the wino is much lighter than the other super-partners. With some assumptions, such a particle realizes the WIMP miracle for a mass around 2 TeV [10]. We proceed by assuming that the correct relic abundance for any mass could in principle be realized by suitable modification of the cosmological history (without diving into the specific details as to how this occurs), and examine the observational constraints on the parameter space based on existing null searches for dark matter.

Our article is organized as follows: Section 2 describes the theoretical framework, including the full set of renormalizable interactions, and the leading higher dimensional operators which lead to splitting of the masses of the states within the electroweak multiplet. Section 3 reviews constraints from high energy accelerators, and Section 4 those from direct searches for the dark matter scattering with terrestrial targets. Section 5 examines the important bounds from indirect searches for dark matter annihilation. We reserve Section 6 for our conclusions.

## 2 Spin Zero SU(2)-Triplet Dark Matter

Our low energy effective theory contains the entire Standard Model plus a real $SU(2)_{EW}$ triplet scalar field $\phi$ with zero hyper-charge. We impose an exact $\mathbb{Z}_2$ discrete symmetry under which the dark matter transforms as $\phi \to -\phi$, and the SM fields are all even, to forbid interactions that could lead to the dark matter decaying into purely SM final states. The most general, renormalizable Lagrangian consistent with these symmetries is:

$$\mathcal{L}_{DM} = \frac{1}{2}(D_\mu \phi)_i (D^\mu \phi)_i - \frac{1}{2}\mu_\phi^2 \phi^2 - \frac{1}{4!}\lambda_\phi \phi^4 - \lambda H^\dagger H \phi^2 \,, \tag{1}$$

where $H$ is the SM Higgs doublet, and $D_\mu \equiv \partial_\mu - i g_w W_\mu^a T^a$ is the gauge covariant derivative with $T_\phi^a$ the generators of $SU(2)_{EW}$ in the triplet representation. The quartic terms whose strengths are parameterized by $\lambda_\phi$ and $\lambda$ characterize the dark matter self-interactions, and an additional connection to the Standard Model via the Higgs portal [11]. Without the $\mathbb{Z}_2$ symmetry, the the term $H^\dagger \phi H$ would be allowed, and would mediate decays through pairs of SM Higgs/Goldstone bosons[1].

The triplet $\phi$ contains a pair of charged fields $\phi^\pm$ and a neutral field $\phi^0$ which plays the role of dark matter. Expanding both $\phi$ and the SM Higgs doublet in components (in the unitary gauge), the Lagrangian density, Eq. (1) reads,

$$
\begin{aligned}
\mathcal{L}_{DM} =\ & \partial_\mu \phi^+ \partial^\mu \phi^- + \frac{1}{2}\partial_\mu \phi^0 \partial^\mu \phi^0 - \frac{\mu_\phi^2}{2}(\phi^0)^2 - \mu_\phi^2 \phi^+ \phi^- \\
& + i g_2 \Big( W_\mu^- (\phi^+ \partial^\mu \phi^0 - \phi^0 \partial^\mu \phi^+) + W_\mu^+ (\phi^0 \partial^\mu \phi^- - \phi^- \partial^\mu \phi^0) \\
& \quad + (A_\mu \sin\theta_W + Z_\mu \cos\theta_W)(\phi^+ \partial^\mu \phi^- + \phi^- \partial^\mu \phi^+) \Big) \\
& + g_2^2 \Big( W_\mu^+ W^{-\mu} \phi^0 \phi^0 + 2 W_\mu^+ W^{-\mu} \phi^+ \phi^- - W_\mu^+ W^{+\mu} \phi^- \phi^- - W_\mu^- W^{-\mu} \phi^+ \phi^+ \\
& \quad + (A_\mu A^\mu \sin^2\theta_W + 2 A_\mu Z^\mu \sin\theta_W \cos\theta_W + Z_\mu Z^\mu \cos^2\theta_W)(\phi^- \phi^+) \\
& \quad - (W_\mu^+ \phi^- \phi^0 + W_\mu^- \phi^+ \phi^0)(A^\mu \sin\theta_W + Z^\mu \cos\theta_W) \Big) \\
& - \lambda \Big( \frac{1}{4}v^2(\phi^0)^2 + \frac{1}{2}vh(\phi^0)^2 + \frac{1}{4}(\phi^0)^2 h^2 + \frac{1}{2}v^2 \phi^+ \phi^- + vh\phi^+ \phi^- + \frac{1}{2}(\phi^+ \phi^-)h^2 \Big) \,.
\end{aligned}
\tag{2}
$$

The interactions with the Standard Model are via the electroweak gauge bosons, whose couplings are controlled by $e$ and $\sin\theta_W$, and take the familiar form dictated by gauge invariance. The interactions with the Higgs boson $h$ are controlled by the Higgs vacuum expectation value (VEV) $v \simeq 246$ GeV and $\lambda$, a free parameter. However, very small values of $\lambda$ represent a fine-tuning, because it is renormalized additively at the one loop level through diagrams such as those shown in Figure 1. These relate the effective value of $\lambda$ at scales $\mu$ and $\mu_0$ (keeping only log-enhanced terms):

$$\lambda(\mu) \simeq \lambda(\mu_0) + \frac{g_2^4}{\pi^2}\ln\left(\frac{\mu^2}{\mu_0^2}\right) \,, \tag{3}$$

which e.g. would induce $\lambda \sim \mathcal{O}(1)$ at the TeV scale if $\lambda$ were taken to vanish at the GUT scale.

---

[1]It is worth noting that Ref. [12] explored a different construction that obviates the need for a $\mathbb{Z}_2$ symmetry by having the dark matter contained in a pseudoscalar triplet that arises from a complex triplet Higgs that mixes through electroweak symmetry-breaking with the SM Higgs doublet.

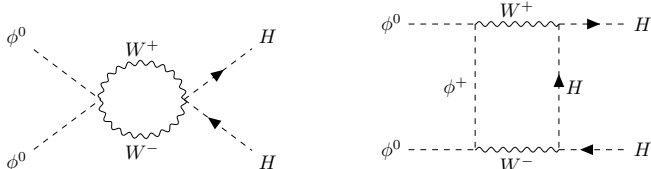

Figure 1: Two representative diagrams contributing to $\lambda$ at one loop.

At tree level, the masses of the charged and neutral components are degenerate, and determined by the parameters $\mu_\phi$ and $\lambda$,

$$m \equiv m_{\phi^0}^2 \;\; = \;\; m_{\phi^\pm}^2 = \mu_\phi^2 + \frac{1}{2}\lambda v^2 \;. \tag{4}$$

At one loop, electroweak symmetry-breaking raises the mass of the charged states, which in the limit of $m \gg v$ results in [13],

$$\Delta m \equiv m_{\phi^\pm} - m_{\phi^0} \approx 166 \,\text{MeV} \;. \tag{5}$$

In the absence of additional ingredients, a strong degeneracy between the masses of the charged and neutral states is inevitable. If one invokes heavy physics which has been integrated out, effectively giving rise to the dimension six operator,

$$\mathcal{L}_{\text{Mass}} = -\frac{1}{\Lambda^2}|\phi^a H^\dagger T^a H|^2 \;\; \rightarrow \;\; -\frac{1}{16\Lambda^2}(\phi^0)^2(v+h)^4 \tag{6}$$

it will shift the mass of the neutral component by

$$\Delta m_{\phi^0}^2 = -\frac{1}{16\Lambda^2}v^4, \tag{7}$$

allowing one to lift the degeneracy by up to $\sim 200$ GeV, for $\Lambda \sim$ TeV. Such an interaction would be induced, for example, by integrating out a mediator SU(2) singlet scalar field $S$ that is odd under the dark $\mathbb{Z}_2$ and has interactions such as $S\phi^a H^\dagger T^a H$. Such a UV completion would be unlikely to further modify the phenomenology we discuss, provided the mass of the $S$ is sufficiently larger than both the mass of $\phi^0$ and the electroweak scale.

The presence of this operator also impacts couplings that can contribute significantly to the rates relevant for direct, indirect, and collider searches, shifting the interactions of $\phi^0$ with one or two Higgs bosons to:

$$\left(\frac{\lambda v}{2} - \frac{v^3}{4\Lambda^2}\right)(\phi^0)^2 h \quad \text{and} \quad \frac{1}{4}\left(\lambda - \frac{3v^2}{2\Lambda^2}\right)(\phi^0)^2 h^2, \tag{8}$$

respectively. We will find it convenient to refer to the strength of the effective $h$-$\phi^0$-$\phi^0$ interaction as $\lambda_{\text{eff}} v/2$, where:

$$\lambda_{\text{eff}} \;\; \equiv \;\; \lambda - \frac{v^2}{2\Lambda^2}. \tag{9}$$

As discussed below, these couplings induce invisible Higgs decays and there are loose constraints on the value of $\Lambda$, which can accommodate mass splittings of up to a few hundred GeV.

# 3 Collider Constraints

The first set of constraints we consider are from the production of dark matter at high energy colliders, such as the LHC and LEP. The rich experimental programs provide multiple complimentary search methods, and probe much of the lower end of the WIMP mass spectrum. Because of the $\mathbb{Z}_2$ symmetry, the underlying production mechanisms in $pp$ or electron-positron collisions involve producing $\phi^0\phi^0$ from Higgs exchange or $W$ boson fusion; $\phi^+\phi^-$ via an intermediate $Z$, $\gamma$, or Higgs boson or from vector boson fusion; and $\phi^0\phi^\pm$ via $W$ exchange or from $W^\pm Z$ fusion. The decay $\phi^\pm \to W^\pm\phi^0$ produces additional SM particles in the final state, which may be very soft when the mass splitting between the charged and neutral states is small. A variety of search strategies attempt to identify distinct signatures from these various DM production channels. Mono-jet searches, invisible Higgs decays, and disappearing charged tracks all apply in different regions of parameter space.

## 3.1 Invisible Higgs Decays

If kinematically allowed $m_\phi \leq M_h/2$, the coupling to the SM Higgs, Eq.(8), allows for Higgs decays into a $\phi^0\phi^0$ final state which escape the detectors ($h \to \text{inv}$), leading to a striking missing energy signal. The irreducible SM background from $h \to ZZ \to 4\nu$, has a branching ratio consistent with the SM expectations $\sim 10^{-3}$ [14] leading to a bound on additional invisible Higgs decay modes, $\mathcal{B}(h \to \text{inv}) \leq 0.19$ [14]. This translates into a bound on a combination of $\lambda$ and $\Lambda$ via the DM contribution to the invisible Higgs decay $h \to \phi^0\phi^0$:

$$\Gamma_{h\to\phi^0\phi^0} = \frac{\sqrt{M_h^2 - 4m_{\phi^0}^2}}{16\pi M_h^2} v^2 \lambda_{\text{eff}}^2, \tag{10}$$

which modifies the Higgs branching ratio into an invisible final state to

$$\mathcal{B}(h \to \text{inv}) = \frac{\Gamma_{\text{DM}}}{\Gamma_{\text{SM}} + \Gamma_{\text{DM}}}. \tag{11}$$

Using the SM Higgs width $\Gamma_{\text{SM}} = 3.2^{+2.8}_{-2.2}$ MeV [15], $\lambda_{\text{eff}}$ must be smaller than

$$\lambda_{\text{eff}} \lesssim (0.102 \text{ GeV}^{1/2}) \times \frac{1}{(M_h^2 - 4m_{\phi^0}^2)^{1/4}} \qquad \left(m_\phi \leq m_H/2\right), \tag{12}$$

which requires $\lambda_{\text{eff}} \lesssim 10^{-2}$ for $m_{\phi^0} \ll M_h$.

## 3.2 Disappearing Tracks

Disappearing charged tracks (DCTs) provide another unique signature. This occurs when a long-lived charged particle hits multiple tracking layers, but disappears due to a decay into a neutral state and a very soft charged particle that escapes detection. In the general context of electroweak multiplet dark matter, if the degeneracy of the masses between the neutral and charged fields are only lifted by electroweak corrections, then decay products may not be able to be detected, and the charged track vanishes at the point of decay.

In the case of the electroweak triplet in the limit of only radiatively-induced mass splitting, the charged states decay $\phi^\pm \to \phi^0\pi^\pm$. Due to the small mass splitting, the lifetime of the charged states is typically long enough for it to hit multiple tracking layers before decaying, and, the momentum of the resulting pion is too soft to be reconstructed, leading to a DCT. For decay rates governed by the electroweak interaction, the LHC is able to rule out this scenario for low $\phi$ masses, requiring [16]:

$$m_{\phi^0} \geq 287 \text{ GeV} \qquad \text{(Compressed Spectrum)}. \tag{13}$$

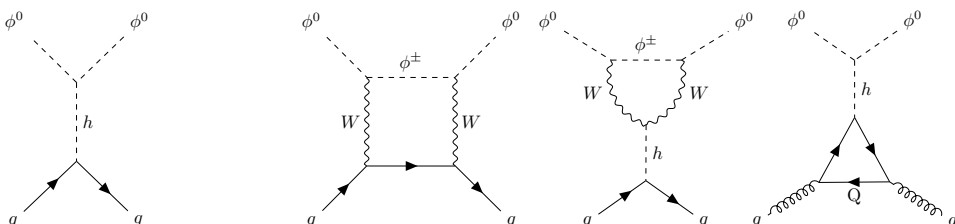

Figure 2: Representative Feynman diagrams for dark matter scattering with quarks/gluons at tree and one loop levels.

### 3.3 Isolated Prompt Leptons

As discussed above, heavy physics may act to induce a mass splitting between the charged and neutral $\phi$ states of up to about 200 GeV. The analysis of Ref. [17] argues that charged partner masses $m_{\phi^{\pm}} \leq 100$ GeV are ruled out for any mass splitting $\Delta m \geq 10$ GeV by a combination of searches at LEP2 combined with LHC results from mono-jet, invisible Higgs decay, and disappearing charged track searches. Generic searches by ATLAS [18, 19] and CMS [20] looking for isolated hard charged leptons are expected to have good sensitivity to production of $\phi^{\pm}$ followed by the decay $\phi^{\pm} \to W^{\pm}\phi^{0}$, but are typically interpreted in the context of specific minimal supersymmetric model parameter points, and are not always trivially recast to apply to the case at hand. Nevertheless, these searches fairly robustly exclude $m_{\phi^{0}} \lesssim 10$ GeV for $m_{\phi^{+}} \lesssim 170$ GeV [18], and a window in $m_{\phi^{+}}$ from $m_{\phi^{+}} \gtrsim (m_{\phi^{0}} + 120$ GeV) to $m_{\phi^{+}} \lesssim 425$ GeV for 20 GeV $\lesssim m_{\phi^{0}} \lesssim 100$ GeV [19]. Thus a moderately compressed spectrum for any dark matter mass above 10-20 GeV, and any uncompressed spectrum with $m_{\phi^{0}} \gtrsim 120$ GeV or $m_{\phi^{+}} \gtrsim 425$ GeV are not constrained by these searches.

## 4 Direct Searches

An important class of constraints on any WIMP come from the null results of searches for the ambient dark matter populating the neighborhood of the Solar System scattering with terrestrial targets. The strongest constraints on WIMPs are typically from experiments searching to detect scattering with heavy nuclei. Given the low expected velocity of Galactic dark matter, the typical momentum transfer is expected to be less than the typical nuclear excitation energies, and the elastic scattering can be described by an effective field theory containing nuclei as degrees of freedom. The nuclear physics is typically unfolded as part of the experimental analysis, and the exclusion limits presented as limits on the spin-independent (SI) or spin-dependent (SD) cross section for scattering with protons or neutrons, extrapolated to zero momentum-transfer [21].

At tree-level, the coupling to the SM occurs through Higgs exchange via $\lambda_{\mathrm{eff}}$, whereas at loop level there are also electroweak contributions [13] (see Figure 2). Integrating out the Higgs and heavy quarks leads to a spin-independent scalar coupling to quarks and gluons,

$$\mathcal{L}_{\mathrm{SI}} = \frac{\lambda_{\mathrm{eff}}\, m_q}{2M_h^2}(\phi^0)^2 \bar{q}q - \frac{\lambda_{\mathrm{eff}}\, \alpha_s}{24\pi M_h^2 v}(\phi^0)^2 G^{\mu\nu}G_{\mu\nu}\,, \tag{14}$$

which are mapped onto an effective coupling to nucleons via the matrix elements [21]:

$$\langle n(k')|m_q\bar{q}q|n(k)\rangle \equiv m_n f_{T,q}^n \bar{u}(k')u(k) \tag{15}$$

$$\langle n(k')|\alpha_s G^{\mu\nu}G_{\mu\nu}|n(k)\rangle \equiv -\frac{8\pi}{9}m_n f_{T,g}^n \bar{u}(k')u(k)\,, \tag{16}$$

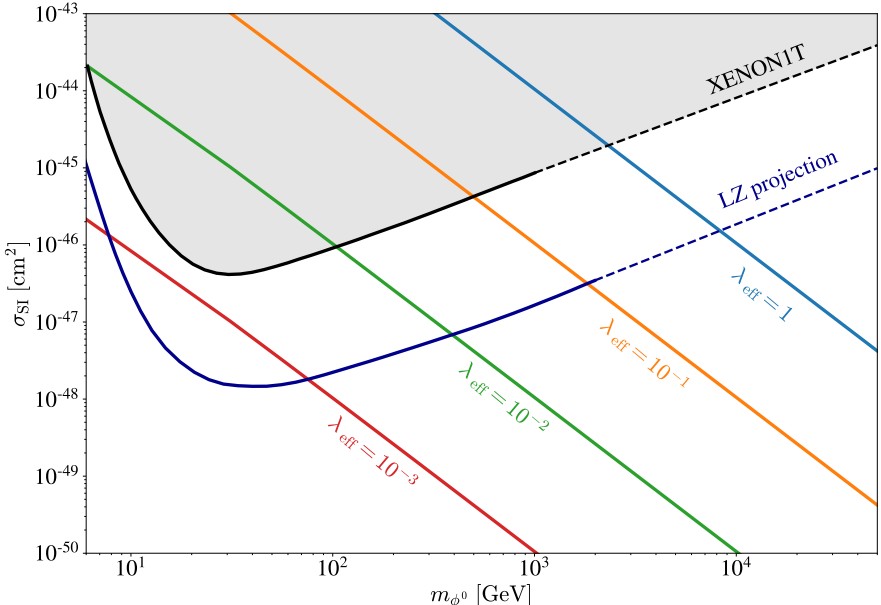

Figure 3: Spin-independent cross section for various values of $\lambda_{\text{eff}}$, as indicated. The current bounds from XENON1T (black) and the projected future sensitivity from LZ (violet) are also indicated.

parameterized by the quantities $f^n_{T,q}$ and $f^n_{T,g}$. The spin independent cross section is

$$\sigma_{\text{SI}}(\phi\,n \to \phi\,n) = \frac{\lambda^2_{\text{eff}}}{4\pi m^2_\phi}\frac{\mu^2_{\phi n}m^2_n}{M^4_h}\left(f^n_{T,u} + f^n_{T,d} + f^n_{T,s} + \frac{2}{9}f^n_{T,g}\right)^2, \tag{17}$$

where $m_n$ is the mass of the nucleon, and $\mu_{\phi n}$ is the reduced mass and the $f^n_T$ approximately satisfy $f^n_{T,u} + f^n_{T,d} + f^n_{T,s} + \frac{2}{9}f^n_{T,g} \approx 0.29$ [22–24]. Because the Higgs coupling is dominated by heavy quarks (contributing through loops to the gluon operator), the scattering with nucleons is approximately isospin symmetric.

Neglecting the small electroweak loop contributions (which, due to partial cancellations, would result in a very small scattering rate of order $10^{-(47-48)}\,\text{cm}^2$ [25] – far below the reach of current direct searches), we show the spin independent cross section as a function of the dark matter mass for several choices of $\lambda_{\text{eff}}$ in Figure 3. Also shown are the current limits on $\sigma_{\text{SI}}$ from the null results of the search for dark matter scattering by XENON1T [26], and projected limits from the LZ experiment [27]. Evident from the figure, XENON1T places an important upper limit on the allowed values of $\lambda_{\text{eff}}$ for a given dark matter mass. To be consistent with any choice of dark matter mass requires,

$$\lambda_{\text{eff}} \lesssim 10^{-3}, \tag{18}$$

with larger values of $\lambda_{\text{eff}}$ permitted for dark matter masses $\gtrsim 30$ GeV or $\lesssim 10$ GeV. Moving forward, we adopt $\lambda_{\text{eff}} \simeq 10^{-3}$ as a benchmark when we discuss indirect searches, below.

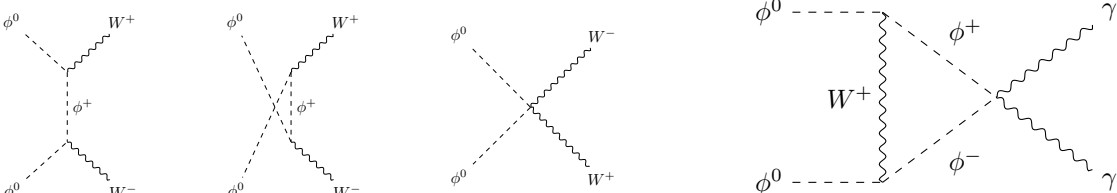

Figure 4: Representative Feynman diagrams for dark matter annihilation into $W^+W^-$ at tree level (left), or into two mono-energetic photons at loop level (right).

## 5  Indirect Searches

Searches for dark matter annihilation in the present day universe provide a complimentary probe of dark matter parameter space, with typical targets including the galactic center (GC) and dwarf spheroidal galaxies (dSphs). We focus on the case of production of high energy gamma rays in dark matter annihilation, which are experimentally accessible over a wide range of energies, and for which the direction of their origin can be measured, providing an additional handle to analyze the dark matter signal. The GC leads to a large potential signal from annihilation, but observations suffer from large astrophysical uncertainties and necessarily complicated regions of interest (RoIs). On the other hand, dSphs provide a lower density source, but generally have much lower backgrounds and uncertainties. Additionally, since there are a variety of dSph Milky Way satellites, stacked analyses can combine the observations to yield stronger and more robust constraints [28].

We consider two important dark matter annihilation processes producing energetic photons: tree level production of continuum photons, and loop level production of mono-energetic gamma ray lines (see Figure 4). At tree level, neglecting the effects of $\lambda_{\text{eff}}$ which we assume for now to be negligibly small, the dark matter can annihilate into a $W^+W^-$ final state that can directly radiate photons; produce them through decays into neutral pions; or produce electrons that radiate via interactions with the interstellar medium and magnetic fields. The predicted gamma ray flux generically depends on the annihilation rate and the distribution of the dark matter within the RoI:

$$\frac{d\Phi}{dE_\gamma}(E_\gamma, \psi) = \frac{\langle \sigma v \rangle}{8\pi m_{\phi^0}^2} \sum_i B_i \frac{dN_i}{dE_\gamma} \times J(\psi), \tag{19}$$

where $\langle \sigma v \rangle$ is the total annihilation cross section; $B_i$ and $dN_i/dE_\gamma$ are the branching fraction and the photon spectrum for final state, $i$, which fully characterize the particle physics information; and

$$J(\psi) \equiv \int d\Omega \int_{\text{los}} \rho^2 ds \tag{20}$$

is the $J$-factor for dark matter annihilation, encoding the information about the density of the dark matter along the line of sight of the observation centered on an angle $\psi$ with respect to the axis from the Earth to the center of the Galaxy.

### 5.1  Annihilation Cross Sections

For dark matter masses above the $W$ mass, the annihilation is dominated by annihilation into on-shell $W^+W^-$, whereas for $M_W/2 \lesssim m_{\phi^0} \leq M_W$ the dominant configuration has one $W$ on-shell, and the other off-shell, and for $m_{\phi^0} \leq M_W/2$, both $W$'s are forced to be off-shell. For $m_{\phi^0} \geq M_W$, the cross section in the zero relative velocity limit reads:

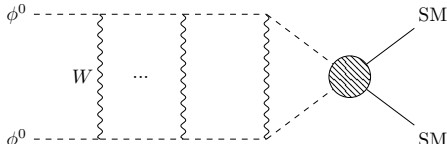

Figure 5: Ladder diagram illustrating the non-perturbative effect of a long-range potential leading to a Sommerfeld-like enhancement.

$$\langle \sigma_{WW} v \rangle = \frac{\sqrt{m_{\phi^0}^2 - M_W^2}}{8\pi m_{\phi^0}^3} \left( g_W^4 \left( \frac{M_W^4}{(M_W^2 - 2m_{\phi^0}^2)^2} + 2 \right) + \frac{6 g_W^2 \lambda_{\text{eff}} M_W^2}{M_h^2 - 4m_{\phi^0}^2} \right.$$
$$\left. + \frac{\lambda_{\text{eff}}^2 (3M_W^4 - 4M_W^2 m_{\phi^0}^2 + 4m_{\phi^0}^4)}{(M_h^2 - 4m_{\phi^0}^2)^2} \right), \tag{21}$$

where the second and third terms may typically be neglected for our benchmark value of $\lambda_{\text{eff}} \sim 10^{-3}$. For dark matter masses below $M_W$, we compute the annihilation into the open final states numerically using the MadDM package [29].

The search for mono-energetic gamma ray lines through $\phi^0 \phi^0 \to \gamma\gamma$ and $\phi^0 \phi^0 \to \gamma Z$ is of particular importance for large dark matter masses in some theories, where the striking signature can balance a suppressed loop-level amplitude [30]. In the limit of large dark matter mass, $m_{\phi^0} \gg M_W$ and $m_{\phi^0} \gg \Delta m$, the cross section for annihilation into $\gamma\gamma$ simplifies considerably, and was computed in Ref. [10] to be

$$\langle \sigma_{\gamma\gamma} v \rangle = \frac{4\pi \alpha_{EM}^2 \alpha_W^2}{M_W^2} \left( 1 + \sqrt{\frac{2\Delta m m_\phi}{M_W^2}} \right)^{-2}, \tag{22}$$

where the last factor provides an important correction at large $m_\phi$ [31]. The rate for annihilation into $\gamma Z$ in the same limit is related by $SU(2)_{\text{EW}}$ gauge invariance [32]:

$$\langle \sigma_{\gamma Z} v \rangle = \frac{6\pi \alpha_{EM} \alpha_W^3}{M_Z^2} \left( 1 + \sqrt{\frac{2\Delta m m_\phi}{M_W^2}} \right)^{-2}. \tag{23}$$

These expressions provide a good qualitative guide to the behavior of the cross sections, but for our quantitative analysis we adopt the calculations of $\phi^0 \phi^0 \to \gamma\gamma$ and $\phi^0 \phi^0 \to \gamma Z$ from Ref. [33], which also contain relevant sub-leading contributions and important re-summation of higher order effects.

## 5.2 Sommerfeld Enhancement

At low velocities, there are potentially important corrections to the annihilation cross section from Sommerfeld-like enhancements originating from formally higher order ladder diagrams such as the one illustrated in Figure 5 [34]. These diagrams encode the additional effective cross section for annihilation due to a long-range attraction between the incoming dark matter particles. The enhanced cross section can be parameterized as

$$\sigma v = S \times \langle \sigma_0 v \rangle, \tag{24}$$

where $\langle \sigma_0 v \rangle$ is the leading order annihilation cross section, and $S$ represents the impact of the Sommerfeld enhancement, given schematically for the case of a strictly massless mediator by

$$S \sim 1 + \frac{\alpha_W}{v} + ... \,. \tag{25}$$

At the low velocities characteristic of the dark matter in the Galaxy ($\sim 10^{-3}$) or in dSphs ($\sim 10^{-5}$), the enhancement for a massless mediator would be a large effect. However, the finite (electroweak size) mediator mass results in a Yukawa potential, for which the Sommerfeld enhancement generically scales more like [35]:

$$S \sim 1 + \alpha_W \frac{m_\phi}{M_W} \,. \tag{26}$$

No closed form expression exists for the Sommerfeld enhancement arising from a Yukawa potential, and we evaluate it numerically. $S$ is strongly depending on both the relative masses of the dark matter and mediator, and the typical velocity of the dark matter. As a result, $\langle \sigma v \rangle$ can differ for e.g. the Galactic center and the dwarf spheroidal galaxies. Where necessary, we provide annihilation cross sections for both, to be compared with the corresponding relevant bound.

### 5.3  $J$ Factor

The $J$-factor depends on the dark matter profile of the source, $\rho(\vec{r})$, which is often not well known. There is wide discussion in the literature concerning which profiles are suggested by data and/or simulations of galaxy formation. Current data is consistent with both cuspy and cored profiles [36, 37]. Pure DM galactic simulations tend to favor cuspy profiles. However, including baryons in simulations provides feedback processes that can smooth the cusps into cores as large as order $\sim$ kpc [38, 39]. An examples of a cuspy and distribution often used in the literature is the Einasto [40] profile given by

$$\rho_{\mathrm{Ein}}(r) = \rho_s \exp\left\{ -\frac{2}{\alpha}\left(\left(\frac{r}{r_s}\right)^\alpha - 1\right)\right\} \,, \tag{27}$$

where $\alpha$ and $r_s$ are parameters typically extracted from simulations [41]. While Einasto is fully consistent with observation, the data also permit profiles with large cores, such as e.g. the Burkert [42] profile, as well.

For small RoIs in the direction of the Galactic Center, the uncertainties in the profile result in a dramatic range of possible $J$ factors, which translate into a wide spread of possible bounds on the annihilation cross section. The H.E.S.S. GC observations place strong constraints on WIMP annihilation when the cuspy Einasto profile is chosen, whereas a cored profile leads to much weaker bounds [31, 43, 44]. This is due in part to the strategy that H.E.S.S. uses to determine its background rate, by comparing a slightly off-center control region (OFF) to the signal (ON) region centered on the Galactic center. As the control region is within about $\sim$ 450 pc of the GC, a $\sim$ kpc sized core would require an accurate extrapolation of the background from the OFF to the ON region to provide a meaningful limit [33]. We simulate cored profiles by assuming an Einasto profile outside of the core radius, with a constant density inside the core. Fortunately, the profiles of dwarf spheroidal galaxies, which are anchored by measurements of stellar kinematics, are much less uncertain than the Galactic center, and thus generally provide more robust constraints [45] (but see also [46]).

### 5.4  Bounds from Indirect Searches

In Fig. 6, we show the predicted annihilation cross section for $\phi^0 \phi^0 \to W^+ W^-$, including the Sommerfeld enhancements expected for the typical dark matter velocity in the Milky Way

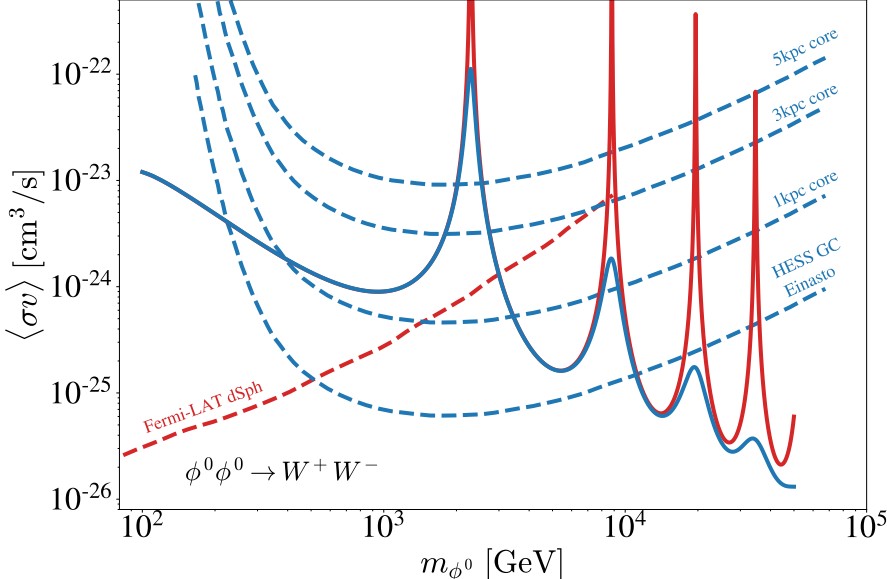

Figure 6: The annihilation cross section for $\phi^0\phi^0 \to W^+W^-$ including the Sommerfeld enhancement with velocities consistent with the galactic center (light blue), and dwarf spheroidal galaxies (red) for comparison with H.E.S.S. (dashed light blue) and Fermi-LAT (dashed red) bounds on this annihilation channel based on gamma rays, respectively.

(MW) Galaxy and in a dwarf spheroidal galaxy. Also shown for comparison are the bounds derived from measurements of gamma rays, interpreted for the $W^+W^-$ final state channel, from the Fermi-LAT observations of dSphs [28] and the H.E.S.S. observations of the GC [47]. For the H.E.S.S. observations, we represent the impact of the uncertainty in the dark matter profile by showing limits for an Einasto profile, as well as for cored profiles with 1 kpc, 3 kpc, and 5 kpc cores. The resulting bounds vary over roughly two orders of magnitude, and for more extremely cored profiles, H.E.S.S. fails to rule out any of the parameter space that is not already excluded by Fermi-LAT[2]. Limits by Fermi from observations of dwarf spheroidals exclude masses from $M_W$ to about $\sim$ 3 TeV. For larger masses, H.E.S.S. extends the region ruled out up to $\sim$ 10 TeV if the profile at the Galactic center is described by Einasto, but little beyond Fermi if the Galactic profile has a core.

As discussed above, for $m_\phi < m_W$, one or both of the W's is forced to go off-shell, leading to more complicated final states including $\phi\phi \to W^\pm ff$ and $\phi\phi \to ffff$. The experimental collaborations limit their presentation of deconvolved bounds to two-body final states, meaning that no careful analysis of the gamma ray spectrum for these final states is readily available from them. To understand the limits below $M_W$, we numerically compute the spectrum of gamma rays for these states using MadDM [29] and use its built-in likelihood analysis to compare with the raw bound on dark matter contributions from the Fermi-LAT observation of the dSphs. The predicted cross section for the inclusive gamma ray spectrum, and the bound derived from it, are shown in Figure 7. In this regime of masses, Fermi-LAT excludes masses in the range 10 GeV $\geq m_\phi \geq$ 60 GeV.

The rates for annihilation into $\gamma\gamma$ and $\gamma Z$ are shown (for Galactic velocities) in Figure 8,

---

[2]It is also worth noting that using the full $\gamma$-ray spectrum from WIMP annihilation, H.E.S.S. excludes a small region around $m_\phi \sim$ 2.3 TeV with a stacked analysis of dSph observations [48].

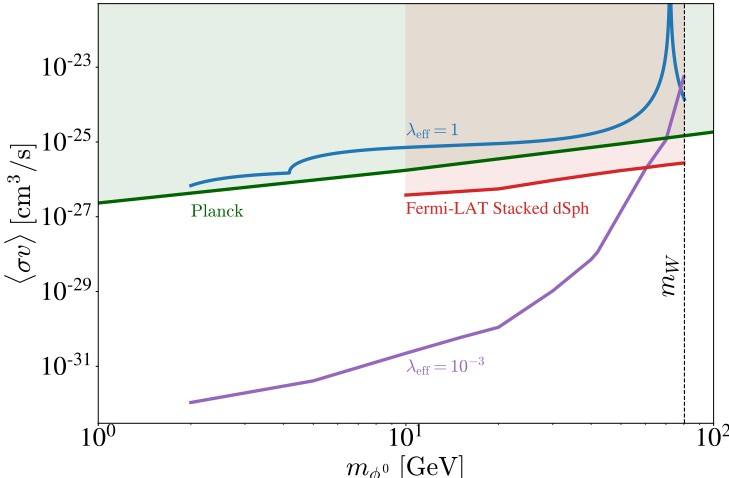

Figure 7: Annihilation cross sections for inclusive gamma ray and positron production when the $\phi^0$ mass is below $M_W$ for $\lambda_{\text{eff}} = 1$ (blue) and $\lambda_{\text{eff}} = 10^{-3}$ (purple), and the corresponding limits on that quantity from Fermi-LAT observations of dwarf spheroidal galaxies (red), derived from the likelihood analysis of MadDM [29], as well as the CMB limit (green) derived using the spectra of $e^+$ and $e^-$ generated by MadDM.

along with the corresponding limits on mono-energetic gamma ray features from H.E.S.S. observations of the inner Galaxy [49] for the Einasto and three differently sized cored profiles. The $\gamma\gamma$ and $\gamma Z$ searches exclude a similar parameter space to the ones derived from annihilation into $WW$ or an Einasto profile, with an additional region probed at the third Sommerfeld resonant peak at $\sim 20$ TeV.

Finally, so far in discussing the bounds from indirect searches we have assumed that $\lambda_{\text{eff}}$ is $\leq 10^{-3}$ to avoid the strong constraints on it from direct searches discussed in Section 4. In order to compete effectively with the annihilation into $W$ pairs, either the mass of the dark matter should be far below $M_W$, or $\lambda_{\text{eff}}$ should be $\gtrsim g_2$, which is consistent with the bounds from XENON1T provided $m_\phi \gtrsim 1.5$ TeV. There is also a tiny region that is resonantly enhanced around $m_\phi \simeq M_h/2$. The Higgs coupling can mediate annihilation to $hh$ for $m_\phi \geq M_h$, or to $\bar{f}f$, dominated by the heaviest fermion kinematically accessible below that. In Figure 9, we show the expected cross sections for annihilations into the two most important channels, $b\bar{b}$ and $\tau^+\tau^-$, for various values of $\lambda_{\text{eff}}$. Comparing with the existing bounds from Fermi and H.E.S.S., it is clear that they do not currently provide additional information beyond the combined requirements of direct searches and limits on annihilation into on-shell or off-shell $W$ bosons.

## 5.5 Constraints from CMB Observables

Precision measurements of the Cosmic Microwave Background (CMB) offer an important vista on dark matter annihilation at late times. A wealth of literature has established the tools to constrain dark matter models using the CMB (see e.g. [50–53]), which can be particularly stringent for light masses. The *Planck* Collaboration provides a robust bound on the annihilation parameter of $f_{\text{eff}}\langle\sigma v\rangle/m_\phi < 4.1 \times 10^{-28} \text{cm}^3/\text{s/GeV}$ [54], where $f_{\text{eff}}$ is the spectrum-weighted efficiency factor [50]. We generate the spectra of $e^+$ and $e^-$ from $\phi\phi$ annihilation into all kinematically accessible two, three, and four-body final states for dark matter masses in the GeV

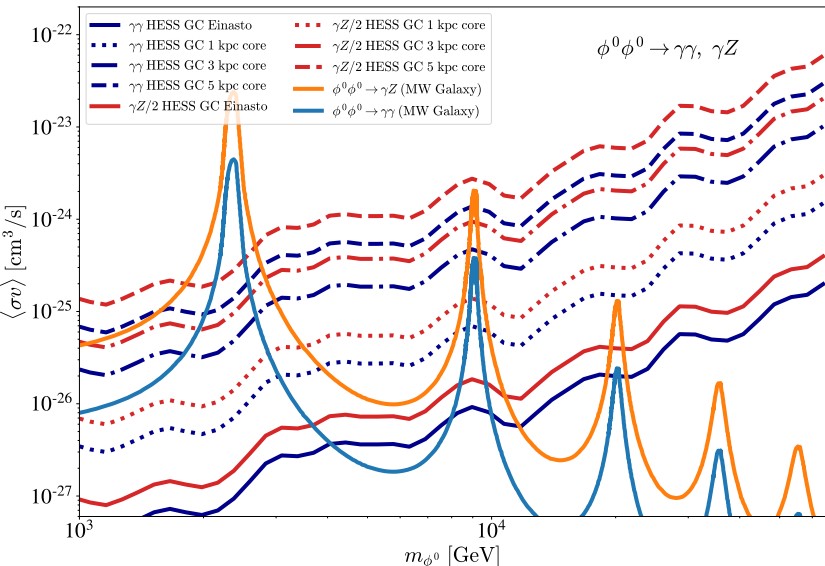

Figure 8: The annihilation cross section for $\phi^0\phi^0 \to \gamma\gamma$ (light blue) and $\phi^0\phi^0 \to \gamma Z$ (orange) based on the re-summed calculation of Ref. [33] for comparison with H.E.S.S. bounds from searches for gamma ray lines for different dark matter profiles (dark blue and red families of curves, respectively.

to TeV range using MadDM [29], from which (following Ref. [50]) we extract the spectrum-weighted $f_{\text{eff}}$. The *Planck* bound thus translates into a bound on the annihilation cross section which is compared to the predicted annihilation cross section in Figure 7. At very low masses, the two-body final states through the Higgs portal dominate the spectrum for all values of the Higgs portal coupling down to $\lambda_{\text{eff}} \simeq 10^{-3}$, and produce CMB bounds that are roughly independent of the values of $\lambda_{\text{eff}}$ we consider. For $\lambda_{\text{eff}} \simeq 1$, the CMB independently excludes masses up to $\sim 700$ GeV. For $\lambda_{\text{eff}} = 10^{-3}$, the CMB excludes $70 \text{ GeV} \lesssim m_\phi \lesssim 500 \text{ GeV}$. In both of these cases, the CMB constraints exclude regions of the parameter space that are also excluded by the other complementary search methods. Nonetheless, since they involve different systematics and are not sensitive to the detailed $J$-factors of astrophysical targets, they provide essential complementary information.

## 6 Conclusions

The question as to the viability of WIMP dark matter remains a subtle one, which in some sense reflects choices as to how to define terms as much as physics. We examine the constraints on real massive scalar particles with full strength electroweak interactions (as triplets) whose abundance in the early Universe is not explicitly tied to standard freeze-out (a situation which may be referred to as a 'miracle-less WIMP'). Such particles very naturally have properties placing them in the right ballpark to play the role of dark matter, without the prejudice on their parameter space implied by the assumption that they froze out during the evolution of a standard cosmology. Their properties are generally captured by three quantities: the mass of the dark matter, the mass of its charged SU(2) sibling, and a dimensionless coupling to the Standard Model Higgs. The strength and form of the inevitable coupling to the electroweak

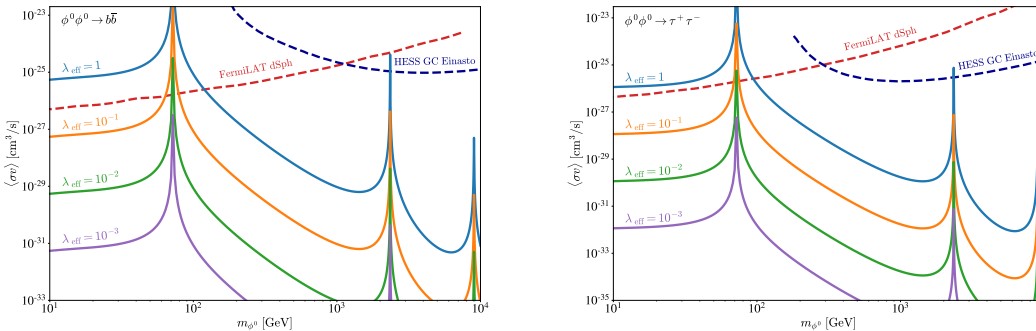

Figure 9: Annihilation cross sections for $\phi^0\phi^0 \to b\bar{b}$ (right panel) and $\phi^0\phi^0 \to \tau^+\tau^-$ (left channel) for different values of $\lambda_{\rm eff}$ as indicated. Also shown are limits on these channels from gamma ray observations by Fermi-LAT (red) and H.E.S.S. (dark blue).

## Is a Miracle-Less WIMP Ruled Out?

Figure 10: Summary of the constraints on an electroweak triplet real scalar field as dark matter.

bosons dictated by gauge invariance is already fixed by the measured SM couplings $e$ and $\sin\theta_W$.

It has been known for some time that the value of the mass for which the WIMP miracle occurs, $m \sim 2$ TeV, is reliably excluded by indirect searches. A summary of the exclusions from various search strategies is presented in Figure 10. Direct searches, often the most stringent constraints on WIMPs, rule out a large range of masses for large values of the Higgs coupling, but are currently unable to say much if this coupling is less than about $10^{-3}$. A Higgs inter-action this small is similarly difficult to discern in rare Higgs decays at the LHC (provided the dark matter is light enough for the Higgs to decay into it) and more direct LHC searches are operative only at very low masses, or for specific large (or tiny) mass splittings between the charged and neutral states. They fairly robustly exclude this scenario for dark matter masses below about 10 GeV, but otherwise can typically be evaded for a moderately compressed spectrum. It is conceivable that a more directed LHC analysis strategy could close the window on a larger swath of the parameter space.

Indirect searches are subject to large uncertainties in the $J$ factor due to our imperfect

knowledge of how dark matter is distributed in astrophysical targets, but do provide key information that does not rely on a large Higgs portal coupling strength. Even for small $\lambda_{\text{eff}}$, a range of masses from around 60 GeV to a few TeV can be reliably excluded by Fermi if the dark matter profile of the Galaxy turns out to have a large core. For a cuspy profile such as Einasto, H.E.S.S. excludes additional parameter space up to around 10 TeV.

Much viable parameter space for a miracle-less scalar electroweak triplet as dark matter remains, albeit constrained in interesting ways which highlight the complementarity of the various search strategies [55]. Our study exemplifies the need for better experimental coverage of the parameter space in order to properly answer the question as to whether simple WIMP models are excluded, or perhaps are present as dark matter but taking an inconvenient incarnation for our current searches.

## Acknowledgements

We thank Graciela Gelmini, Rocky Kolb, Simona Murgia, Manoj Kaplinghat, Anyes Taffard, Mauro Valli, and especially Tracy Slatyer for helpful discussions. This work is supported in part by U.S. National Science Foundation Grant No. PHY-1915005.

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
