# Peer review of "Is a Miracle-less WIMP Ruled Out?"

_SciPost Physics, doi:SciPost Phys. 11, 019 (2021)_

## Round 1 · Referee Report · Anonymous · 2021-6-24

Strengths

1) The paper studies motivated physics beyond the standard model

2) The indirect detection analysis is very detailed

Weaknesses

1) The introduction of the higher-dimensional operator does not seem strongly motivated to be and it has a clear impact on the phenomenology

Report

This paper studies a dark matter model where the candidate is a complex scalar transforming as a triplet under the weak isospin. The authors introduce the theoretical framework in section 2. There are three different classes of interactions: couplings with gauge bosons, quartic Higgs portal in the scalar potential, and the dimension 6 effective operator given in equation (6). The last one makes it possible to lift the degeneracy between neutral and charged states, which is only due to gauge boson loop corrections at the renormalizable level. The authors consider the usual bounds on weak scale dark matter: collider (section 3), direct detection (section 4), indirect detection (section 5). They ignore relic density constraints that would otherwise exclude the model, and this is justified by the fact that dark matter freeze-out takes place way earlier than BBN and modified cosmological histories altering the relic density prediction are allowed. A nice summary of this study can be found in figure 10 where all the experimental bounds are accounted for.

The paper reads very well and everything is explained in a clear way. I believe the results are interesting enough to deserve to appear in the literature and I am happy to recommend this paper for publication.

---

## Editorial Decision

published